# On the Unintended Social Bias of Training Language Generation Models with News Articles

## Abstract

There are concerns that neural language models may preserve some of the stereotypes of the underlying societies that generate the large corpora needed to train these models. For example, gender bias is a significant problem when generating text, and its unintended memorization could impact the user experience of many applications (e.g., the smart-compose feature in Gmail).

In this paper, we introduce a novel architecture that decouples the representation learning of a neural model from its memory management role. This architecture allows us to update a memory module with an equal ratio across gender types addressing biased correlations directly in the latent space. We experimentally show that our approach can mitigate the gender bias amplification in the automatic generation of articles news while providing similar perplexity values when extending the Sequence2Sequence architecture.

## 1 Introduction

Neural Networks have proven to be useful for automating tasks such as question answering, system response, and language generation considering large textual datasets. In learning systems, bias can be defined as the negative consequences derived by the implicit association of patterns that occur in a high-dimensional space. In dialogue systems, these patterns represent associations between word embeddings that can be measured by a Cosine distance to observe male- and female-related analogies that resemble the gender stereotypes of the real world. We propose an automatic technique to mitigate bias in language generation models based on the use of an external memory in which word embeddings are associated to gender information, and they can be sparsely updated based on content-based lookup.

The main contributions of our work are the following:

- We introduce a novel architecture that considers the notion of a Fair Region to update a subset of the trainable parameters of a Memory Network.

- We experimentally show that this architecture leads to mitigate gender bias amplification in the automatic generation of text when extending the Sequence2Sequence model.

## 2 Memory Networks and Fair Region

As illustrated in Figure 1, the memory $M$ consists of arrays $K$ and $V$ that store addressable *keys* (latent representations of the input) and *values* (class labels), respectively as in Kaiser et al. (2017). To support our technique, we extend this definition with an array $G$ that stores the *gender* associated to each word, e.g., *actor* is *male*, *actress* is *female*, and *scientist* is *no-gender*. The final form of the memory module is as follows:

$$M = (K, V, G).$$

A neural encoder with trainable parameters $\theta$ receives an observation $x$ and generates activations $h$ in a hidden layer. We want to store a normalized $h$ (i.e., $\|h\| = 1$) in the long-term memory module $M$

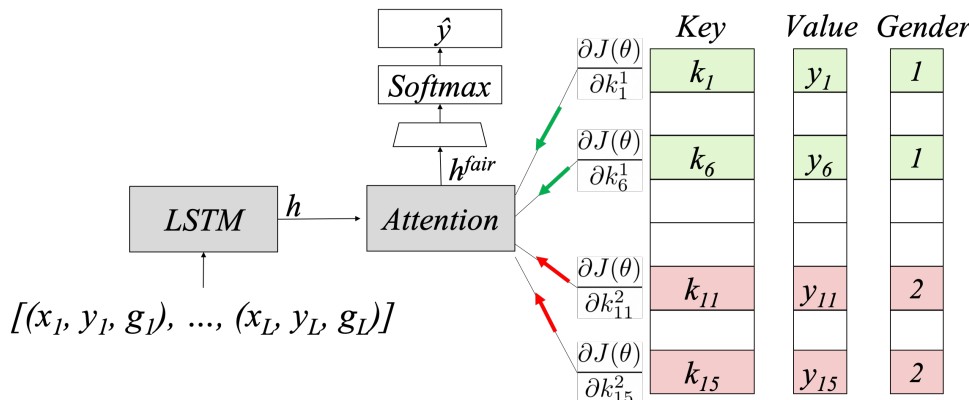

Figure 1: A Fair Region in memory $M$ consists of the most similar keys to $h$ given a uniform distribution over genders (e.g., 1: *male*, and 2: *female*). The input consists of a sequence tokens annotated with gender information, e.g., (*The*, 0), (*president*, 0), (*gave*, 0), (*her*, 2), (*speech*, 0).

to increase the capacity of the encode. Hence, let $i_{max}$ be the index of the most similar key

$$i_{max} = argmax_i\{h \cdot K[i]\},$$

then writing the triplet $(x, y, g)$ to $M$ consist of:

$$K[i_{max}] = \|h + K[i_{max}]\|$$
$$V[i_{max}] = y$$
$$G[i_{max}] = g.$$

However, the number of word embeddings does not provide an equal representation across gender types because context-sensitive embeddings are severely biased in natural language, Zhao et al. (2017). For example, it has been shown in that *man* is closer to *programmer* than *woman*, Bolukbasi et al. (2016). Similar problems have been recently observed in popular work embedding algorithms such as Word2Vec, Glove, and BERT, Kurita (2019).

We propose the update of a memory network within a Fair Region in which we can control the number of keys associated to each particular gender. We define this region as follows.

**Definition 2.1. (Fair Region)** Let $h$ be an latent representation of the input and $M$ be an external memory. The *male*-neighborhood of $h$ is represented by the indices of the $n$-nearest keys to $h$ in decreasing order that share the same gender type *male* as $\{i_1^m, ..., i_k^m\} = KNN(h, n, male)$. Repeating the same process for each gender type estimates the indices $i^f$ and $i^{ng}$ resulting in the *female* and *non-gender* neighborhoods. Then, the *FairRegion* of $M$ given $h$ consists of $K[i^m; i^f; i^{ng}]$.

The Fair Region of a memory network consists of a subset of the memory keys which are responsible for computing error signals and generating gradients that will flow through the entire architecture with backpropagation. We do not want to attend over all the memory entries but explicitly induce a uniform gender distribution within this region. The result is a training process in which gender-related embeddings equally contribute in number to the update of the entire architecture. This embedding-level constraint prevents the unconstrained learning of correlations between a latent vector $h$ and similar memory entries in $M$ directly in the latent space considering explicit gender indicators.

## 3 LANGUAGE MODEL GENERATION

Our goal is to leverage the addressable keys of a memory augmented neural network and the notion of fair regions discussed in Section2 to guide the automatic generation of text. Given an encoder-decoder architecture Sutskever, Vinyals, and Le (2014); Bahdanau, Cho, and Bengio (2015), the inputs are two sentences $x$ and $y$ from the source and target domain, respectively. An LSTM encoder outputs the context-sensitive hidden representation $h^{enco}$ based on the history of sentences and an LSTM decoder

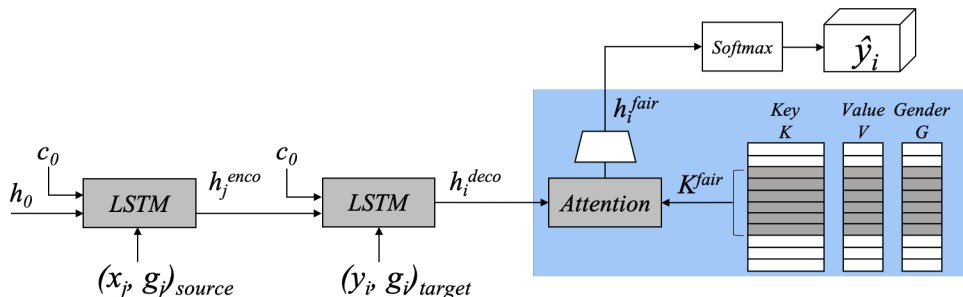

Figure 2: Architecture of the neural dialogue model that incorporates the memory approach discussed in Section 1. The figure shows the $i^{th}$ decoding step of the word $\hat{y}_i$ given the sparse update within a Fair Region centered at $h^{deco}$.

receives both $h^{enco}$ and $y$ and predicts the sequence of words $\hat{y}$. At every timestep of decoding, the decoder predicts the $i^{th}$ token of the output $\hat{y}$ by computing its corresponding hidden state $h_i^{deco}$ applying the recurrence

$$h_i^{deco} = LSTM(y_{i-1}, h_{i-1}^{deco}).$$

Instead of using the decoder output $h_i^{deco}$ to directly predict the next word as a prediction over the vocabulary $O$, as in Miller et al. (2016). We combine this vector with a query to the memory module to compute the embedding vector $h_i^{fair}$. We do this by computing an attention score Bahdanau, Cho, and Bengio (2015) with each key of a Fair Region. The attention logits become the unnormalized probabilities of including their associated values for predicting the $i^{th}$ token of the response $\hat{y}$. We then argmax the most likely entry in the output vocabulary $O$ to obtain the $i^{th}$ predicted token of the response $\hat{y}$. More formally,

$$K^{fair} = FairRegion(h^{deco}, K, n)$$
$$\alpha_i = Softmax(h_i^{deco} \cdot K^{fair})$$
$$h_i^{fair} = Wtanh(\alpha_i \cdot K^{fair})$$
$$\hat{y} = Softmax(h_i^{fair})$$
$$\hat{y}_i = O[argmax_j(\hat{y}[j])].$$

Naturally, the objective function is to minimize the cross entropy of actual and generated content:

$$J(\theta) = -\sum_{j=1}^{N}\sum_{i=1}^{m} y_i^j \, log \, p(\hat{y}_i^j)$$

where $N$ is the number of training documents, $m$ indicates the number of words in the generated output, and $y_i^j$ is the one-hot representation of the $i^{th}$ word in the target sequence.

## 4 BIAS AMPLIFICATION

Inspired by Zhao et al. (2017), we compute the bias score of a word $x$ considering its word embedding $h^{fair}(x)$[1] and two gender indicators (words *man* and *woman*). For example, the bias score of *scientist* is:

$$b(scientist, man) = \frac{\left\|h^{fair}(scientist)\right\| \cdot \left\|h^{fair}(man)\right\|}{\left\|h^{fair}(scientist) \cdot h^{fair}(man) + h^{fair}(scientist) \cdot h^{fair}(woman)\right\|}.$$

If the bias score during testing is greater than the one during training,

$$b^{test}(scientist, man) - b^{train}(scientist, man) > 0,$$

then the bias of *man* towards *scientist* has been amplified by the model while learning such representation, given training and testing datasets similarly distributed.

---

[1] For Seq2Seq neural models, this word embedding is the output of the decoder component $h^{deco}(x)$

| | PERPLEXITY | | | BIAS AMPLIFICATION | | |
|---|---|---|---|---|---|---|
| MODEL | ALL | PERU | MEXICO | ALL | PERU-MEXICO | MEXICO-PERU |
| SEQ2SEQ | 13.27 | 15.31 | 15.61 | +0.18 | +0.25 | +0.21 |
| SEQ2SEQ+ATTENTION | **10.73** | 13.25 | 14.08 | +0.25 | +0.32 | +0.29 |
| SEQSEQ+FAIRREGION | 10.79 | **13.04** | **13.91** | **+0.09** | **+0.17** | **+0.15** |

Table 1: Perplexity and Bias Amplification results on the datasets of crawled newspapers.

## 5 EXPERIMENTS

### 5.1 DATASET

We evaluate our proposed method in datasets crawled from the websites of three newspapers from Chile, Peru, and Mexico.

To enable a fair comparison, we limit the number of articles for each dataset to 20,000 and the size of the vocabulary to the 18,000 most common words. Datasets are split into 60%, 20%, and 20% for training, validation, and testing. We want to see if there are correlations showing stereotypes across different nations. *Does the biased correlations learned by an encoder transfer to the decoder considering word sequences from different countries?*

### 5.2 BASELINES

We compare our approach **Seq2Seq+FairRegion**, an encoder-decoder architecture augmented with a Fair Region, with the following baseline models:

1. **Seq2Seq** Sutskever, Vinyals, and Le (2014): An encoder-decoder architecture that maps between sequences with minimal assumptions on the sequence structure and that is able to remember long term dependencies by mapping the source sentence into a fixed-length vector.

2. **Seq2Seq+Attention** Bahdanau, Cho, and Bengio (2015): Similar to Seq2Seq, this architecture automatically attends to parts of the input that can be relevant to predict the target word.

### 5.3 TRAINING SETTINGS

For all the experiments, the size of the word embeddings is 256. The encoders and decoders are bidirectional LSTMs of 2-layers with state size of 256 for each direction. For the Seq2Seq+FairRegion model, the number of memory entries is 1,000. We train all models with Adam optimizer Kingma and Ba (2014) with a learning rate of 0.001 and initialized all weights from a uniform distribution in $[-0.01, 0.01]$. We also applied dropout Srivastava et al. (2014) with keep probability of $95.0\%$ for the inputs and outputs of recurrent neural networks.

### 5.4 FAIR REGION RESULTS IN SIMILAR PERPLEXITY

We evaluate all the models with test *perplexity*, which is the exponential of the loss. We report in Table 1 the average perplexity of the aggregated dataset from Peru, Mexico, and Chile, and also from specific countries.

Our main finding is that our approach (Seq2Seq+FairRegion) shows similar perplexity values (10.79) than the Seq2Seq+Attention baseline model (10.73) when generating word sequences despite using the Fair Region strategy. These results encourage the use of a controlled region as an automatic technique that maintains the efficacy of generating text. We observe a larger perplexity for country-based datasets, likely because of their smaller training datasets.

## 5.5 FAIR REGION CONTROLS BIAS AMPLIFICATION

We compute the *bias amplification* metric for all models, as defined in Section 4, to study the effect of amplifying potential bias in text for different language generation models.

Table 1 shows that using Fair Regions is the most effective method to mitigate bias amplification when combining all the datasets (+0.09). Instead, both Seq2Seq (+0.18) and Seq2Seq+Attention (+0.25) amplify gender bias for the same corpus. Interestingly, feeding the encoders with news articles from different countries decreases the advantage of using a Fair Region and also amplifies more bias across all the models. In fact, training the encoder with news from Peru has, in general, a larger bias amplification than training it with news from Mexico. This could have many implications and be a product of the writing style or transferred social bias across different countries. We take its world-wide study as future work.

## 6 CONCLUSIONS

Gender bias is an important problem when generating text. Not only smart composer or auto-complete solutions can be impacted by the encoder-decoder architecture, but the unintended harm made by these algorithms could impact the user experience in many applications. We also show the notion of bias amplification applied to this dataset and results on how bias can be transferred between country-specific datasets in the encoder-decoder architecture.

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
