# OpenReview forum: "On the Unintended Social Bias of Training Language Generation Models with News Articles"
_ICLR.cc/2020/Conference — Reject_

### Official Review · AnonReviewer3 · 2019-10-22
**Official Blind Review #3**

**Rating:** 1

**Review:**


Summary:
The authors propose using attention with Fair Region in Memory Network for reducing gender bias in language generation.

Decision:
Overall, the paper is poorly written and lack of enough substance, even though the idea seems reasonable. I am inclined to reject this paper.

Supporting argument:
1. The authors use quite small dataset (also non-standard) for evaluation. It might be better to evaluate on some standard large-scale language model datasets.
2. As mentioned in  Section 5.5, the proposed approach is not generalized to other countries.
3. Since the task is language generation, the use of bias-amplification in embedding space is not well justified.
4. What are the words used in the Fair Region?
5. In section 3, the V, G of memory are not used. What's the output of FairRegion function?

Additional feedback:
1. The usage of notation, citation is quite poor.
- The usage of \citep and \cite are different.
- In Table 1, what are PERU-MEXICO, MEXICO-PERU models?
- Section2 -> Section 2

**Experience Assessment:**

I have read many papers in this area.

**Review Assessment: Checking Correctness Of Derivations And Theory:**

N/A

**Review Assessment: Checking Correctness Of Experiments:**

I assessed the sensibility of the experiments.

**Review Assessment: Thoroughness In Paper Reading:**

I made a quick assessment of this paper.

---

### Official Review · AnonReviewer1 · 2019-10-23
**Official Blind Review #1**

**Rating:** 3

**Review:**

This paper focuses on the timely and important topic of unintended social bias in language models and proposes an approach to mitigate gender bias in the automatic generation of news articles. The automatic text generation is done by a sequence-to-sequence-model using word embeddings. The pattern of these embeddings in an multi-dimensional space can be associated with the stereotypes found in the corpora the models are trained on. To mitigate this bias the authors combine the network with a memory module storing additional gender information. This approach is inspired by the memory module used by Kaiser et al. (2017) to remember rare events, replacing the vector to track the age of items with a key representing the gender.
The proposed network architecture ("Seq2Seq+FairRegion") is evaluated using the bias score presented by Zhao et al. (2017) and compared to the sequence-to-sequende-models "Seq2Seq2" by Sutskever, Vinyals and Lee (2014) and "Seq2Seq+Attention" by Bahdanau, Cho and Bengio (2015).

Besides some typographical errors ("work embedding") the paper is overall well written but unfortunately the flow of the text could be improved to make it easier to follow. The authors should perhaps consider to explain the structure of the paper in the introduction. There are some errors according the brackets of the in-text citations which should be corrected.
The two figures both illustrate the memory module and could be easily combined to one because the second figure just adds the encoding stage of the architecture. The caption of figure 1 could be used to further explain the "Fair Region" in the text (section 2). In section 4 the authors explain how to compute the bias score. Because these calculations are used to evaluate their methods, this section should perhaps be a subsection to section 5 ("experiments").
To evaluate the proposed method the authors use newspaper articles from Chile, Peru and Mexiko and compare the bias amplification metric for the trained models. The results are shown for the datasets from Peru and Mexico (Chile is missing) with a larger bias amplification for the Peru dataset. It could be justified more clearly why the experiments compare the bias amplification scores of different countries while the main focus of the paper is to mitigate gender bias in general.

While this is an important topic, the contribution is rather minor and not well presented. Specifically the experiements should be extended.

**Experience Assessment:**

I have read many papers in this area.

**Review Assessment: Checking Correctness Of Derivations And Theory:**

I assessed the sensibility of the derivations and theory.

**Review Assessment: Checking Correctness Of Experiments:**

I assessed the sensibility of the experiments.

**Review Assessment: Thoroughness In Paper Reading:**

I read the paper at least twice and used my best judgement in assessing the paper.

---

### Official Review · AnonReviewer2 · 2019-10-24
**Official Blind Review #2**

**Rating:** 1

**Review:**

The paper considers the problem of debiasing word representation in a trained language model using some sort of key-value memory structure with annotations of gender that is constrained to view only a subset of keys.

The current form of the manuscript contains too few details to evaluate the work in a meaningful way. Please include

(1) a more detailed description of data input and output for system (i.e. where are gendered bits coming from), and more details of the memory (is it a global fixed memory? then how gendered bits assigned?)

(2) a more complete definition of bias amplification -- beyond just an example (i.e. what training representation are you comparing to?). A formula would be useful here.

(3) more complete experimental definition. What datasets exactly are you evaluating on?

(4) overview sections to motivate your solutions and your setup (i.e. an introduction, related work)

**Experience Assessment:**

I have published in this field for several years.

**Review Assessment: Checking Correctness Of Derivations And Theory:**

N/A

**Review Assessment: Checking Correctness Of Experiments:**

N/A

**Review Assessment: Thoroughness In Paper Reading:**

I read the paper at least twice and used my best judgement in assessing the paper.

---

### Decision · Program_Chairs · 2019-12-19

**Decision:**

Reject

**Comment:**

The reviewers had a hard time fully identifying the intended contribution behind this paper, and raised concerns that suggest that the experimental results are not sufficient to justify any substantial contribution with the level of certainty that would warrant publication at a top venue. The authors have not responded, and the concerns are serious, so I have no choice but to reject this paper despite its potentially valuable topic.